# Improving Rice Pest Management Through RP11: A Scientifically Annotated Dataset for Adult Insect Recognition

**DOI:** 10.3390/life15060910

**Published:** 2025-06-04

**Authors:** Biao Ding, Yunxiang Tian, Xiaojun Guo, Longshen Wang, Xiaolin Tian

**Affiliations:** 1School of Electronic and Information Engineering, Faculty of Innovation Engineering, Macau University of Science and Technology, Macau 999078, China; 1210019409@student.must.edu.mo; 2School of Computer Science and Engineering, Faculty of Innovation Engineering, Macau University of Science and Technology, Macau 999078, China; 2109853gii30009@student.must.edu.mo; 3Department of Plant Protection, College of Plant Protection, Shanxi Agricultural University, Taiyuan 030031, China; guoxj0302@126.com; 4Department of Artificial Intelligence, College of Computer Science and Technology, Huaqiao University, Xiamen 361000, China; 2325191027@stu.hqu.edu.cn

**Keywords:** rice pest, deep learning, dataset, YOLOv11

## Abstract

Rice yields are expected to drop significantly due to the increasing spread of rice pests. Detecting rice pests in a timely manner using deep learning models has become a prevalent approach for rapid pest control. However, current datasets related to rice pests often suffer from limited sample sizes or poorly annotated labels, which compromises the training accuracy of deep learning models. Building upon the large-scale IP102 dataset, this study refines the rice pest segment of IP102 by separating adult specimens and larva specimens, acquiring additional pest images via web crawler techniques, and re-annotating all adult samples. The pest category names, originally in English, are replaced with the Latin scientific names of the corresponding families to improve both clarity and scientific accuracy. The resulting dataset, designated RP11, includes 11 adult categories with 4559 images and 7 larval categories with 2467 images. All annotations follow a labeling format compatible with YOLO model training. The sample count in RP11 is approximately four times that of the rice-specific subset in IP102. In this work, YOLOv11 was employed to evaluate RP11’s performance, with IP102 serving as a comparison dataset. The results demonstrate that RP11 outperforms IP102 in precision (83.0% vs. 58.9%), recall (79.7% vs. 63.1%), F1-score (81.3% vs. 60.9%), mAP50 (87.2% vs. 62.0%), and mAP50–95 (73.3% vs. 37.9%).

## 1. Introduction

Rice is one of the most widely consumed staple crops globally. In China, for instance, approximately 60% of the population relies on rice as their primary food source [1]. According to the World Food and Agriculture Organization (FAO), global rice production reached 799.9 billion kg in 2023, with China contributing 208 billion kg, accounting for roughly 26% of the total output. Rice is also a key agricultural product in many other countries [2,3,4,5,6]. The types of rice pests vary by region. For example, rice planthoppers (family *Delphacidae*), stem borers, and leaf folders are among the most damaging pests in China’s rice-producing regions [1], whereas the rice water weevil (family *Curculionidae*) is considered the most destructive pest in the United States [7]. Rice crops affected by pests can suffer production losses ranging from 30% to 40% [6]. To reduce pest-related yield loss, the use of pesticides remains the most common strategy. It is estimated that chemical pesticides have prevented around one-third of potential yield losses [8]. In practice, a single insecticide is often used against pests from the same family or even the same order [9,10,11]. However, excessive pesticide use may result in the development of pest resistance and increase the risk of poisoning incidents [6]. Thus, fast and accurate identification of pest species is essential for effective pest control in agriculture.

With the trend toward large-scale and automated agricultural cultivation, alongside rapid advancements in machine learning and deep learning, researchers have increasingly recognized the alignment between these technologies and agricultural needs, leading to practical applications. Wang Jinsheng et al. designed the RP-DETR model by integrating modules like Gold-YOLO and Transformer [12] and adopting improvements such as the MPDIoU loss function, achieving significant performance gains over RT18-DETR and RT34-DETR [13]. Liangquan Jia’s team developed MobileNet-CA-YOLO, an enhanced YOLOv7 model based on MobileNet and attention mechanisms, which attained 92.3% accuracy and 93.7% mAP50 in detecting three common rice diseases and three rice pests [14]. Zheng Yong et al. improved YOLOv8n into Rice-YOLO by replacing the detection head with DBG and introducing Aux-DBG for mid-level supervision, achieving 78.1% mAP50, 62.9% mAP50–95, and 74.3% F1 scores for detecting 16 rice pests, outperforming the original YOLOv8n [15]. Yin Jiandong and Zhu Jun proposed YOLO-RMD, a modified YOLOv8n architecture tailored for detecting seven common rice pests, achieving an average precision of 98.2% [16]. These studies highlight the effectiveness of optimized deep learning models in advancing precision pest and disease detection in agriculture.

The growing demand for accurate pest identification in agriculture has driven the adoption of machine learning, particularly for rice pest detection, where models like the YOLO series [17] have shown significant success [18]. The YOLO series of models and its series of improved models have become the most frequently used models in the field of rice pest and disease detection [14,15,16,19,20]. The YOLO family has evolved through continuous innovation: YOLOv3 [21] introduced multi-scale prediction and Darknet-53 for small object detection; YOLOv4 [22] added mosaic augmentation and CSPDarknet53; YOLOv5 [23] streamlined training procedures; YOLOv6 [24] and YOLOv7 [25] optimized deployment efficiency; YOLOv8 [26] enhanced feature fusion; YOLOv9 [27] incorporated self-supervised learning and hybrid attention; YOLOv10 [28] utilized neural architecture search for edge computing; YOLOv11 [29] balanced speed and accuracy via dynamic backbones and multi-task optimization; and the latest version, YOLOv12 [30], harmonized attention mechanisms with CNN efficiency, achieving state-of-the-art real-time detection performance. These advancements highlight the YOLO series’ role in advancing real-time object detection for agricultural applications.

Beyond model design, dataset quality and scale critically determine deep learning performance. While early rice pest datasets were limited in size (e.g., Alfarisy’s 4511-image dataset covering nine pests and four diseases [31]), IP102 has emerged as the dominant benchmark with 75,222 images of 102 pest species, including 8417 rice pest images across 14 categories (1249 XML-annotated) [32]. We noticed that the dataset IP102 was selected for model training in a large number of studies [13,15,16,33] and is the most widely used dataset for pests. Subsequent improvements led to IP_RicePest—a version of IP102’s rice-specific data enhanced through rigorous screening and augmentation, now containing 14,000 images [33]. Notably, the unannotated IPR16 dataset offers over 5000 rice pest images but lacks labels for supervised learning [15].

To address the issues of unbalanced sample distribution, insufficient rigor, low accuracy, and unsuitable labeling format for YOLO model training in the most widely used rice pest dataset, we conducted screening, expansion, and re-annotation, resulting in a new dataset named RP11. This dataset includes 18 categories (11 species pest in total), organized by common rice pest families, and comprises 4559 image samples. We then evaluated performance using the YOLOv11 model and conducted a comparative training experiment with the rice portion of IP102.

## 2. Related Work

Datasets form the foundation for training deep learning models. Supervised learning generally yields more accurate outcomes than unsupervised learning [34]. As such, datasets with high-quality samples and precise annotations are essential for effective deep learning training.

Jiangning Wang, Chongtian Lin, Liqiang Ji, and Aiping Liang created a dataset categorized by order, covering a total of 10 insect orders, with 225 images in total [35]. Chenjun Xie and colleagues developed a dataset with 24 species, containing around 60 images per species [36]. As a subfield of machine learning, deep learning models (such as YOLO) typically require more training samples than traditional models like SVM or KNN. Datasets with limited image counts may fall short of supporting such models. Z. Liu et al. compiled a dataset featuring 12 common pest classes and approximately 5000 images. They employed a DCNN model and adjusted the images so that the pest was centered in each frame. However, none of the images in this dataset were annotated [37].

IP102 is a large dataset for insect pest recognition, containing 75,222 images across 102 categories [32]. It includes eight super-classes: ‘Rice’, ‘Corn’, ‘Wheat’, ‘Beet’, ‘Alfalfa’, ‘Vitis’, ‘Citrus’, and ‘Mango’. The dataset was tested using the CNN model ResNet [38], which delivered the best performance. Each category in IP102 comprises multiple life stages of the pest, contributing to the dataset’s overall coverage. Among these, the classification accuracy for rice pests was the lowest, at 32.1%. IP_RicePests is a dataset derived from the rice pest portion of IP102 that was filtered and expanded [33]. A key shortcoming of both datasets lies in their use of English names for classification, which has led to sample misclassifications and presents challenges for practical application. We chose to focus on the rice subset, which showed the lowest accuracy in IP102. Furthermore, only a small portion of the samples in IP102’s rice pest section, 1249 across 14 classes, are labeled, which is not ideal for machine learning or deep learning models that rely on supervised learning. Addressing this limitation was one of the goals of our work.

Zhiyong Li et al. developed a dataset named IP_RicePest based on IP102 [33]. They addressed the issue of limited sample size by collecting additional images and applying the Attentive Recurrent Generative Adversarial Network (ARGAN) [39] for sample enhancement, resulting in a dataset with 14,000 images. However, the dataset still inherits some of the same problems as IP102, such as the lack of rigor due to classification by English names and variability in sample quality. These are among the issues our study aims to resolve.

The IRP16 [15] dataset was established by Zheng Yong’s team, constructed by integrating their previously developed R2000 dataset (containing 2024 images of 16 rice pest species) with relevant rice pest samples from the IP102 dataset. This merged collection resulted in a comprehensive dataset comprising 2556 images covering 16 distinct rice pest species.

The three most critical datasets in this study are IP102, IP_RicePest, and IPR16. IP_RicePest primarily refines IP102 by filtering and augmenting its 14 pest categories while retaining IP102’s classification framework. IPR16 incorporates IP102 as a data source, selecting 532 images (547 instances) from it and expanding the categories from 14 to 16 by integrating data from previous studies. While both datasets improve upon IP102, they retain its English common names for pest classification, which may cause regional misinterpretations due to language differences. Since using Latin scientific names for classification is widely recognized as the most precise and universal approach, we further expanded and filtered images to establish the RP11 dataset based on Latin nomenclature.

## 3. Materials and Methods

### 3.1. Basis of Classification

In current pest recognition datasets, a common approach involves grouping different developmental stages of the same pest species under a single taxonomic label. A notable example is the IP102 dataset, where categories such as “Yellow rice borer” and “Asiatic rice borer” include multiple life stages, adults of *Chilo suppressalis* (Walker) and *Scirpophaga incertulas* (Walker), their larvae, egg masses, and even symptoms on plants resulting from infestation (as shown in Figure 1). While this broad inclusion helps expand dataset coverage for training deep learning models capable of recognizing pests across life stages, it leads to biological inaccuracies due to limited input from entomological specialists during dataset development.

From a biological standpoint, insect development presents two fundamentally distinct immature forms: larvae, which are typical of complete metamorphosis, and nymphs, associated with incomplete metamorphosis. Complete metamorphosis (e.g., *Lepidoptera*, *Coleoptera*) involves larval forms that are morphologically, physiologically, and ecologically distinct from adults, often inhabiting different niches and exhibiting different feeding behaviors. In contrast, incomplete metamorphosis (e.g., *Hemiptera*, *Orthoptera*) involves nymphs that closely resemble adults in form and behavior, sharing ecological niches and resources throughout development.

Integrating taxonomic expertise with deep learning frameworks presents a unique interdisciplinary challenge. Our core objective was to construct scientifically rigorous datasets that satisfy both taxonomic precision and deep learning operational requirements, thereby ensuring the reliability and deployability of trained models. To bridge these domains, we invited a plant conservation specialist to participate. This expert oversaw dataset curation and participated in multiple rounds of screening of the images involved in the dataset, enforcing strict adherence to taxonomic protocols while maintaining algorithmic compatibility. The collaborative workflow guaranteed that the final datasets are both biologically authentic and computationally tractable, enabling the development of models that balance scientific validity with industrial applicability.

To bridge this disconnect between biological structure and computational modeling, we propose a metamorphosis-aware taxonomic strategy:•Segmented classification: Separate larval and adult stages for holometabolous species into distinct classes.•Unified classification: Maintain a single class for hemimetabolous species, where feature continuity across nymphs and adults is preserved.

This framework aligns the dataset’s taxonomic structure with entomological accuracy while improving the consistency of learned features. Importantly, our approach prioritizes interdisciplinary collaboration, incorporating detailed entomological validation through partnerships with agricultural entomologists to ensure the dataset remains biologically sound.

### 3.2. Identify Classifications

Rice, a staple crop vital to global food security, faces serious threats from pest infestations that significantly reduce yields [6]. Accurately identifying pest species and estimating their population densities in the field remains a challenge. This has prompted growing interest in applying computer vision techniques integrated with deep learning models for fast and reliable pest detection. To meet this demand, we developed RP11, a dedicated dataset centered on rice pests, categorized by taxonomic family.

The dataset includes eleven arthropod families, selected based on their agricultural relevance as primary predators of rice crops. Among them, seven families undergo holometabolous development, as described in Section 3.1. For classification, we used Latin scientific names instead of English common names to ensure taxonomic precision, since vernacular terms often lack specificity and can result in misclassification [9,10,11].

This approach is consistent with real-world pest management, where pesticides effective against one species are often also effective within the same family [9,10,11]. In addition, closely related species tend to share morphological traits, which benefits machine learning-based recognition. As outlined in Table 1, the dataset structure captures these biological relationships and includes the following:•Adult specimens from all 11 families (visual examples in Figure 2).•Larval stages from the seven holometabolous families (as shown in Figure 3).

### 3.3. Data Acquisition

To obtain specimen images with validated taxonomic accuracy, we employed a dual-source data collection approach:•Primary acquisition from the established IP102 benchmark dataset, emphasizing its peer-reviewed pest imagery repository.•Supplementary collection through systematic retrieval from verified entomological image databases under Creative Commons licensing.

#### 3.3.1. Data from IP102

We carried out a structured screening of pest categories within the IP102 dataset using a multi-stage selection framework. As all selected families include species known to damage rice crops, the process began by identifying 14 pest categories relevant to rice based on their documented agricultural significance [32]. To maintain high data quality, we implemented a strict image assessment protocol.

Visual inspection was conducted independently by professional agricultural personnel, following predefined criteria:•Key morphological features clearly visible.•No severe occlusion or distracting background elements.

Larval specimens were recognized using morphological indicators such as the following:•Body proportions characteristic of larval forms.•Distinctive traits consistent with the given family.

Adult specimens were validated by the following:•Adult-stage body proportions.•Diagnostic features including wing venation and antennal structures.

This selection process achieved three primary outcomes:•Improved intra-class consistency through quality filtering.•Accurate developmental stage labeling to support lifecycle-specific recognition.•Alignment with entomological taxonomic standards.

Table 2 lists the 14 rice-related pest species and their corresponding families as classified in IP102. Using this method, we acquired approximately 2500 adult images and 2000 larval images.

#### 3.3.2. Data from Image Databases

To maintain taxonomic accuracy while respecting intellectual property rights, our image acquisition process was based on scientific name-specific searches from sources compliant with copyright regulations. Using ethical web-scraping techniques with controlled parameters, we obtained more than 3000 images from these families. These images were initially screened by us, and the ones of lower quality were removed (as shown in Figure 4). Subsequently, the experts in the field of plant protection within our project team conducted a secondary confirmation. After this step, we collected 2150 licensed specimen images from 11 families, all obtained under public domain licenses. These pictures were placed in their respective categories according to the family names used during the search.

To increase intraspecific diversity without compromising taxonomic clarity, we deliberately expanded specimen representation within each family. This family-level expansion strategy draws on the biological principle that morphological similarity tends to be greater among species within the same family than across distantly related taxa. For example, our *Curculionidae* category includes multiple congeneric species such as *Lissorhoptrus oryzophilus* (a major rice pest), *Sitophilus zeamais* (Motschulsky), *Myllocerinus aurolineatus* (Voss), and *Sitophilus oryzae* (Linnaeus). Figure 5 provides visual comparisons, highlighting their shared family-level traits.

#### 3.3.3. Pre-Treatment of Images

The species-to-family taxonomic alignments and nomenclatural accuracy detailed in Section 3.3.1 were rigorously validated by an expert specializing in plant protection, ensuring congruence with current taxonomic frameworks (e.g., International Code of Zoological Nomenclature). Furthermore, the morphological fidelity of images presented in Section 3.3.2 was critically verified to confirm their unambiguous attribution to the respective insect families within the classification system, utilizing comparative morphological diagnostics and reference-specimen cross-referencing. This dual-layer validation protocol adheres to peer-reviewed standards for taxonomic reliability and minimizes misidentification risks in biodiversity studies.

The biological images assigned to the same family in Section 3.3.1 and Section 3.3.2 were systematically consolidated into unified taxonomic categories. For holometabolous species (i.e., those exhibiting complete metamorphosis with distinct larval stages), these categories were further subdivided into adult and larval subcategories. Plant protection taxonomists, adhering to standardized morphological criteria (e.g., presence/absence of sclerotized wings, genitalia development, and cephalic capsule differentiation in larvae), rigorously segregated larval and adult stage images into their respective subcategories. This hierarchical partitioning aligns with the International Code of Zoological Nomenclature.

#### 3.3.4. Dataset Overview

The final compiled dataset contains 7026 taxonomically validated entomological specimens. These are systematically divided into 4559 adult samples across 11 taxonomic families and 2467 larval samples from 7 holometabolous families. Figure 6 displays the sample count distribution for each adult category in the RP11 dataset and Figure 7 shows the distribution of the sample number for each larval category in the RP11 dataset.

#### 3.3.5. Image Naming Rules

To improve dataset clarity and usability, we implemented a structured naming protocol, applying separate coding systems for adult and larval specimens. Adult specimen names follow the format RPxxyyy, where “xx” indicates the taxonomic family code and “yyy” represents the specimen’s serial number within that category. Larval specimens use the format LRPaabbbb, with “aa” as the family identifier and “bbbb” as the image number.

This dual-coding structure supports taxonomic consistency and transparency, allowing for immediate recognition of the developmental stage and the number of samples per category. For example, RP05005 identifies the fifth adult image in family 05, while LRP090033 refers to the thirty-third larval image in family 09. Importantly, corresponding developmental stages within the same family share the same numeric family code, “xx” for adults and “aa” for larvae, ensuring alignment across stages. For instance, specimens from Curculionidae are consistently labeled RP01yyy for adults (e.g., RP01217 refers to adult image number 217) and LRP01bbbb for larvae (e.g., LRP010121 refers to larval image number 121), maintaining traceability within the family.

### 3.4. Deep Learning Model Selection

The growing demand for accurate pest identification in agriculture has spurred significant advancements in machine learning applications, particularly through the evolution of the YOLO series [17]. Our statistical analysis of the existing literature reveals that the YOLO series is the most widely adopted deep learning framework in rice pest detection research, and newer iterations like YOLOv12 continue to emerge. Despite the rapid evolution of the YOLO architecture, earlier versions (e.g., YOLOv5 to YOLOv8 [14,15,16]) demonstrate enduring relevance through strategic modular enhancements tailored to agricultural contexts. Researchers frequently achieve state-of-the-art performance by retrofitting proven components such as attention mechanisms, advanced feature fusion networks, or lightweight backbones into legacy YOLO frameworks. This adaptability is particularly valuable in agricultural computing environments with resource constraints, where lightweight variants of YOLOv5 or YOLOv8 often outperform newer models in speed–accuracy tradeoffs when optimized for edge deployment.

Initially popularized for real-time object detection, YOLO’s continuous innovation has made it indispensable for agricultural pest recognition. Early versions like YOLOv3 [21] addressed critical challenges in small pest detection through multi-scale prediction and the Darknet-53 backbone, while YOLOv4 [22] enhanced robustness with mosaic augmentation and CSPDarknet53. Subsequent iterations such as YOLOv5 [23] streamlined training workflows, lowering the barrier for agricultural adoption, and YOLOv6 [24]/YOLOv7 [25] optimized deployment efficiency through architectural refinements. The series reached new milestones with YOLOv8’s [26] advanced feature fusion techniques and YOLOv9’s [27] integration of self-supervised learning and hybrid attention mechanisms, enabling finer-grained pest discrimination. YOLOv10 [28] further tailored solutions for edge computing in field environments using neural architecture search, while YOLOv11 [29] introduced dynamic backbone networks and multi-task optimization to balance speed and precision. The latest iteration, YOLOv12 [30], marks a paradigm shift by harmonizing attention mechanisms with CNN efficiency, achieving state-of-the-art accuracy without compromising real-time performance—a critical requirement for time-sensitive pest management. These advancements have been validated across numerous studies, where YOLO variants demonstrated exceptional capabilities in detecting rice pests like stem borers, leaf rollers, and planthoppers, even under complex field conditions involving occlusions and varying illumination. By automating pest identification with subsecond inference speeds and high precision, YOLO-based systems reduce reliance on labor-intensive manual scouting, enable early intervention, and support precision pesticide application. Ongoing research focuses on enhancing model adaptability to diverse pest morphologies and environmental contexts, leveraging synthetic data augmentation and cross-domain transfer learning to address data scarcity challenges in agricultural settings.

Current trends show iterative optimization of YOLOv10/v11 architectures for multi-pest recognition systems, leveraging dynamic neural networks to handle rice ecosystems’ biodiversity. These findings underscore YOLO’s unique position in agricultural AI—its version-agnostic modularity allows continuous performance gains without requiring complete architectural overhauls, ensuring backward compatibility, crucial for real-world deployment.

In the current research on rice pest detection, the application of YOLOv11 is still a blank frontier. This study aims to construct a benchmark experimental framework for YOLOv11: on the one hand, to verify its basic performance through an unimproved native model; on the other hand, to establish a standardized evaluation benchmark in combination with a newly constructed high-quality dataset. This work not only fills the application gap of YOLOv11 in the field of agricultural detection, but also generates three basic data—the performance baseline of the native model, the data quality impact coefficient, and the reference for dataset improvement comparison—providing a traceable experimental control group for subsequent research and facilitating the exploration of the optimization direction of the new dynamic network architecture in agricultural scenarios.

### 3.5. Professional Data Annotation

As detailed in Section 3.3.4, both adult and larval classifications exhibit a pronounced long-tail distribution, with the imbalance particularly evident in the larval categories. This distribution reflects real-world entomological patterns and ecological constraints. For instance, the dominance of Crambidae larvae (n = 1266) corresponds with their major role as agricultural pests, especially as stem borers, whose larvae create feeding tunnels within rice stems, a damage trait that is easily detected during field observations. This is further linked to the broad geographic distribution of major *Crambidae* species, notably *Chilo suppressalis* and *Scirpophaga incertulas*. By contrast, the rarity of *Hydrellia philippina* larvae (n = 4) mirrors their specialized habitat, submerged root zones, which poses significant challenges for photographic collection.

In light of these constraints, our annotation strategy emphasizes taxonomic labeling for adult specimens while treating larval categories as a distinct hierarchical layer. This phased method incorporates two key provisions:•Provisional larval taxonomy preservation: Current larval categories are retained with biologically valid classification, supplemented by dormant taxonomic metadata.•Conditional re-annotation trigger: Systematic relabeling of larval categories begins when two conditions are met: (1) sample count reaches 500 per taxon, and (2) ecological coverage reaches 85% across endemic regions.

To support supervised learning applications, we applied a structured annotation protocol across the dataset. Each image specimen was annotated using YOLO-compliant .txt label files [17], adhering to the following normalized format:<class_id><x_center><y_center><width><height>

Our annotation process followed rigorous quality control protocols. Two annotators adhered to standardized guidelines to create precise bounding boxes that fully encompassed biological features while tightly fitting specimen contours. The YOLO format’s coordinate-based simplicity facilitated consistent labeling across annotators. All coordinates were scaled to fall within the [0,1] range relative to image dimensions (illustrated in Figure 8). After initial annotation, all labels underwent dual verification: first by the original annotator for positional accuracy; then, by plant protection experts, confirming biological validity through morphological and ecological authenticity checks.

### 3.6. Evaluation Indicators

We evaluated the dataset using precision, recall, mAP50, and mAP50–95. Precision measures the proportion of correctly predicted positive samples within a category, while recall indicates the proportion of actual positive samples correctly detected by the model. Intersection over union (IoU) is calculated as the ratio between the overlapping area and the total area covered by the predicted and actual bounding boxes. It is used to assess the alignment between predicted and ground truth boxes. mAP50 represents the mean average precision at an IoU threshold of 50%, while mAP50–95 averages precision across IoU thresholds ranging from 50% to 95% with a step of 5%.(1)$$Precision=\frac{TP}{TP+FP}$$(2)$$Recall=\frac{TP}{TP+FN}$$(3)$$F1-score=\frac{2\times Precision\times Recall}{Precision+Recall}$$(4)$$mAP50=\frac{AP50_{1}+AP50_{2}+\ldots +AP50_{N}}{N}$$(5)$$mAP50–95=\frac{mAP50+mAP55+\ldots +mAP95}{10}$$
where TP is the number of correctly predicted positive samples, TN is the number of correctly predicted negative samples, FP refers to positive samples that were incorrectly predicted, and FN refers to negative samples that were incorrectly predicted. $AP50_{1}$ represents the average precision for the first category at an IoU threshold of 50%. The same method applies to the other categories. N denotes the total number of categories.

In particular, we set an indicator AP50–95 to evaluate the multi-threshold performance on a single classification: (6)$$AP50–95_{n}=\frac{AP50_{n}+AP55_{n}+\ldots +AP95_{n}}{10}$$
where n is one of all the categories. This indicator helps to evaluate a particular category in detail.

The relationship between mAP50–95 and AP50–95 can be obtained from the formula for calculating mAP50–95 (Formula (5)).(7)$$\begin{array}{cc} mAP50–95 & =\frac{mAP50+mAP55+\ldots +mAP95}{10}\\ \; & =\frac{\frac{AP50_{1}+AP50_{2}+\ldots +AP50_{N}}{N}+\frac{AP55_{1}+AP55_{2}+\ldots +AP55_{N}}{N}+\ldots +\frac{AP95_{1}+AP95_{2}+\ldots +AP95_{N}}{N}}{10}\\ \; & =\frac{\frac{\left(AP50_{1}+AP55_{1}+\ldots +AP95_{1}\right)+\left(AP50_{2}+AP55_{2}+\ldots +AP95_{2}\right)+\ldots +\left(AP50_{N}+AP55_{N}+\ldots +AP95_{N}\right)}{N}}{10}\\ \; & =\frac{\frac{\left(AP50_{1}+AP55_{1}+\ldots +AP95_{1}\right)+\left(AP50_{2}+AP55_{2}+\ldots +AP95_{2}\right)\ldots +\left(AP50_{N}+AP55_{N}+\ldots +AP95_{N}\right)}{10}}{N}\\ \; & =\frac{\left(\frac{AP50_{1}+AP55_{1}+\ldots +AP95_{1}}{10}\right)+\left(\frac{AP50_{2}+AP55_{2}+\ldots +AP95_{2}}{10}\right)+\left(\frac{AP50_{N}+AP55_{N}+\ldots +AP95_{N}}{10}\right)}{N}\\ \; & =\frac{AP50–95_{1}+AP50–95_{2}+\ldots +AP50–95_{N}}{N} \end{array}$$

For k-fold cross-validation, we need to determine the stability of the dataset and the model. We need to obtain the mean $\bar{X}$ and variance $\sigma^{2}$ of k experiments and calculate the confidence interval based on them.(8)$$\bar{X}=\frac{1}{n}\sum_{i=1}^{n}X_{i}$$(9)$$\sigma^{2}=\frac{1}{n}\sum_{i=1}^{n}{\left(X_{i}-\bar{X}\right)}^{2}$$(10)$$95\%\;confidence interval=\bar{X}\pm t_{0.025,n-1}\times \frac{\sigma }{\sqrt{n}}$$
where n is the number of iterations.

### 3.7. Training Environment

All experiments were conducted on a machine running Windows 11, equipped with an AMD Ryzen 7 7735H CPU, 16 GB of RAM, and an NVIDIA RTX4060 GPU. The software environment was based on Python 3.8.20.

YOLOv11 was selected as the deep learning model for training. Table 3 presents the YOLOv11 parameter settings used in the experiment.

## 4. Results

Deep learning performance metrics serve as key indicators for assessing dataset quality. In this section, we evaluate the adult categories in the RP11 dataset using the YOLOv11 model. The model was configured and trained according to the parameters outlined in Section 3.7. Metrics such as precision, recall, and mAP were used to provide a comprehensive evaluation of RP11’s performance. As a control, we also trained YOLOv11 using the 14 rice pest categories from IP102 with the same parameter settings.

### 4.1. Verification of Dataset Stability

k-fold cross-validation serves as a critical methodology for evaluating model stability, wherein multiple train–validation splits maximize data utilization while mitigating data underutilization and random bias inherent to single splits. Our protocol first reserves 10% of the entire dataset as an independent test set, then divides the remaining 90% into nine equal folds (k = 9). For each iteration, one fold serves as the validation set and the remaining eight as the training set, with all splits rigorously enforcing taxonomic-level stratified sampling to preserve class distribution integrity. Aligned with the training parameter standards outlined in Section 3.6, we conducted nine experimental runs, recording precision, recall, mAP50, mAP50–95, and F1-score metrics. The overall mean and variance were calculated using Formulas (8) and (9), from which we derived the 95% confidence intervals (Formula (10)) for each metric, statistically quantifying performance consistency across all taxonomic groups.

The results of the nine iterations are shown in Table 4.

$t_{0.025,n-1}$ = 2.306 when n = 9. According to the formula, we can obtain the calculation results shown in Table 5.

### 4.2. Performance of RP11

The dataset was partitioned into training, validation, and test sets at an approximate ratio of 8:1:1, with stratification strictly implemented at the taxonomic/classification level to preserve hierarchical representation integrity. The detailed numerical distributions across the subsets are shown in Table 6.

We conducted a full evaluation of YOLOv11’s performance on RP11. After 100 training epochs, performance metrics were obtained for both the overall dataset and each individual category (summarized in Table 7).

The evaluation metrics (precision, recall, mAP50, mAP50–95, F1-score) showed the following results: recall, mAP50, and mAP50–95 fell within the 95% confidence interval of k-fold cross-validation. Precision was 2.6% below the interval’s lower bound, and F1-score—calculated from precision and recall—was 0.5% below the lower bound, both within normal fluctuations. These findings confirm the reliability of the experimental data. According to Table 7, *Hesperiidae* showed the strongest performance among all categories, with the highest precision (100%), the highest recall (97.5%), the highest AP50 (99.0%), the second-highest AP50–95 (85.7%), and the highest F1-score (98.7%).

In contrast, *Phlaeothripidae* and *Ephydridae* performed less effectively on F1-score.

For *Phlaeothripidae*, it shows high precision (83.5%) and low recall (44.1%). This reflects the practical challenges in detecting Thripidae (thrips) species in object detection systems: despite their distinct biological characteristics, their extremely small body size (typically ≤2 mm) and high-density clustering patterns (particularly flower-attracted males) create unique computational hurdles. In field imagery, these pests often manifest as clusters of dozens of dim, near-identical micro-particles against complex backgrounds—a scenario where current detection frameworks struggle with three key issues:•Scale–sensitivity conflict—their minute size falls below standard detection thresholds even in high-resolution (≥4K) images;•Feature ambiguity—limited pixel occupancy (often <10 × 10 pixels) prevents effective texture/shape feature extraction;•Occlusion noise—overlapping individuals in dense groups create pseudo-macrostructures that confuse detection heads.

For *Ephydridae*, it shows a high recall and a low precision, which means there are fewer missed detections and more false detections. The reason for this situation is that the sample size of *Ephydridae* is relatively small. This is a practical problem arising from the actual situation. Because the distribution of *Ephydridae* in the real ecological environment is small and its size is relatively small (about 3 mm), its features are relatively blurred at a wide angle or a longer viewing angle, making identification difficult.

### 4.3. Actual Effect Display

In this study, only adult specimens were annotated, while larval stages were excluded. Although biologically, adults and larvae represent different developmental stages of the same species, from a computer vision perspective, taxonomic classification relies on shared visual features. Larval and adult forms typically lack overlapping morphological characteristics, making it more challenging for models to recognize them as a single class than to classify adult specimens from different species within the same taxonomic family. In our dataset, larval stages were categorized separately but excluded from annotation. This approach avoids the challenge of requiring models to identify shared morphological features between adults and larvae, thereby simplifying training. However, the lack of sufficient larval specimens to support model training renders larval detection infeasible, ultimately limiting the practical applicability of the research.

As shown in Figure 9, when larvae of a pest appear in an image or video frame, they are not detected, but adults of the same species are correctly identified.

When a large number of pest instances are present in an image or video frame, the model can still perform accurate identification (as shown in Figure 10).

Most of the instances on the left-hand side of the image were correctly identified, while the samples in the middle and on the right were not. This indicates that when the edges or features of the samples are blurred, the recognition ability of the model is weak (such as the samples in the middle part of the image), and when the color of the samples deviates significantly from the real situation, the recognition difficulty will also be higher (such as the samples on the right-hand side of the image).

In scenarios where multiple pest species appear in a single image or video frame, the model is still capable of accurate recognition, as demonstrated in Figure 11.

When objects occupy few pixels or overlap in images or video frames, the model’s detection performance significantly degrades, leading to missed or false detections. This situation is more obvious in small size categories (as shown in Figure 12) and similar categories (as shown in Figure 13).

### 4.4. Comparison of RP11 and IP102

In order to demonstrate the performance of the RP11 dataset, we selected the rice pest part of IP102, one of the most widely used datasets, as the dataset used in our comparison experiment.

#### 4.4.1. Number of Samples

IP102 is one of the important sources of image samples in our dataset. To allow direct comparison, we used IP102 for a controlled experiment. Table 8 provides a comparison of the number of annotated images between RP11 and IP102. We followed the same partitioning method used in IP102, dividing RP11 into training, validation, and test sets at a ratio of 8:1:1. The sample counts for each subset in both datasets are shown in Table 9.

#### 4.4.2. Confusion Matrix and the Normalization

The confusion matrix is a standard tool used to evaluate the performance of classification models. It compares actual labels with predicted labels, identifying how many samples were correctly or incorrectly classified (e.g., true positives, false positives). This matrix provides a clear view of how the model performs across different categories and supports the calculation of key metrics such as accuracy and recall. To address the impact of imbalanced class distributions, the normalized confusion matrix expresses these values as proportions. Normalizing by rows reflects recall across categories, while normalizing by columns reflects the accuracy of the predicted outputs. This method helps reduce the bias caused by uneven sample numbers and allows for more balanced comparisons between categories. It is particularly useful for identifying categories that are frequently confused with each other and for refining classification strategies.

Using both standard and normalized confusion matrices allows for deeper analysis of specific error patterns while also evaluating the model’s overall balance and reliability. In the matrix, the horizontal axis (columns) represents the model’s predicted class, while the vertical axis (rows) represents the actual class. Each cell shows the number of samples predicted as a particular class. The closer the values along the diagonal are to the maximum, the better the model is at correctly classifying that class. Lower values off the diagonal indicate misclassifications. After training, the results of the confusion matrices and normalized confusion matrices of RP11 and IP102 are shown in Figure 14 and Figure 15.

From the values in Figure 14 and the proportions in Figure 15, as well as the color depth representing their proportions in the two figures, we can clearly find that in the confusion matrix obtained according to RP11’s training, the number of samples correctly predicted is far greater than the number of samples incorrectly predicted, which reflects the high precision of the training results. The column shows that the number of successfully identified positive samples is also much higher than the number of positive samples that are not identified, reflecting the high recall.

#### 4.4.3. Precision, Recall, and F1-Score

Precision, recall, and F1-score are three important indexes to evaluate the training results of the model. Precision and recall represent two different perspectives of evaluation, while F1-score reflects a comprehensive index obtained as the *harmonic mean* of precision and recall.

•Precision indicates the proportion of correctly predicted positive samples among all samples which are actually positive (as shown in Formula (1)). Higher precision means fewer false positives. In the actual pest protection situation, high precision and low false alarm rate are very important. Because the wrong judgment of the model will lead to use of the wrong pesticide types and dosages in agricultural production, which will lead to serious economic losses.•Recall measures how many actual positive samples are correctly identified by the model, reflecting its detection coverage (as shown in Formula (2)). A higher recall means more real (positive) samples are covered by the testing process and successfully identified.•Formula 3 shows the calculation process of the F1-score. The formula is a variant formula of the harmonic mean formula. Instead of simply finding the average of the two, the F1-score needs to take into account the difference between precision and recall. When one of the indicators goes up and the other goes down, the F1-score shows a downward trend. What this achieves in the actual test is that when F1-scores drop, it means that there is a certain imbalance between precision and recall. This forces the system to capture as many real objects as possible while maintaining a high accuracy rate, avoiding either extreme.

The results for precision, recall, and F1-score are presented in Table 10. Figure 16, Figure 17 and Figure 18 provide visual comparisons between RP11 and IP102. Among them, Figure 16 and Figure 17 show the variation in precision and recall under different confidence thresholds. Confidence refers to the degree to which the model is certain of the predicted results. For example, when the confidence threshold is set to 70%, the model will only predict that samples with certainty greater than 70% are positive samples. This means that at higher confidence thresholds, precision is higher and recall is lower. In actual deployment, the user can adjust the confidence threshold based on the current need for high precision or high recall. It should be noted that the F1-score listed in the table represents the theoretical maximum (81.3%) based on precision and recall under ideal conditions, while the highest F1-score shown in Figure 18 reflects the actual performance. For IP102, when the confidence threshold is set to 0.268, the model achieves its highest overall F1-score, reaching 53%. For RP11, when the confidence threshold is set to 0.489, the model achieves its highest overall F1-score, reaching 80%.

#### 4.4.4. Precision–Recall Curve

The precision–recall curve is a valuable tool for assessing classification performance, particularly when category distributions are imbalanced. It illustrates the relationship between precision and recall across varying classification thresholds: increasing the threshold typically raises precision but lowers recall, and vice versa.

The area under the curve serves as an overall performance indicator, the closer the curve approaches the top-right corner of the plot (value near 1), the better the model’s ability to maintain both high precision and high recall. Figure 19 displays the precision–recall curves for RP11 and IP102.

#### 4.4.5. mAP50 and mAP50–95

The mAP (mean Average precision) is the core evaluation indicator in the field of object detection, measuring the degree of coincidence between the predicted frame and the real frame by the IoU. The calculation process integrates the integral area of the precision–recall curve (PR curve), which can fully reflect the comprehensive performance of the model under different confidence thresholds.

•mAP50: When the IoU threshold is set to 50% (that is, the overlap area between the predicted box and the real box is greater than 50%), the average precision (AP) of all categories is calculated, and then the average value is taken. A tolerance assessment that allows for “rough positioning”, mAP50 mainly evaluates whether the model detects the target, and the requirements for positioning accuracy are relaxed.•mAP50–95: Under a progressively strict standard for the IoU threshold, from 50% to 95% (usually with 5% as a step, a total of 10 thresholds), the AP values are calculated separately and then averaged. This is equivalent to a step test from “identification presence” to “pixel-level positioning”.

Table 11 shows the results of mAP50 and MAP50–95 obtained by training YOLOv11 using RP11 and the rice pest part of IP102.

## 5. Discussion

In this study, RP11 comprises 11 adult categories with 4559 images and 7 larval categories with 2467 images. The dataset exhibits a natural long-tail distribution (as shown in Figure 6 and Figure 7), meaning some categories contain many samples while others have very few. This imbalance is a common phenomenon and, to a certain extent, reflects the actual severity and prevalence of these pests in real-world agricultural environments. For example, in the United States, the most serious rice pest is the rice water weevil (*Lissorhoptrus oryzophilus*) [7], making *Curculionidae* one of the most populous categories in RP11. Likewise, *Delphacidae* are major rice pests across the globe [40,41], especially in Asia [42,43] and Africa [44]. Their sample count in RP11 is second only to that of *Curculionidae*. To take another example, the larvae of *Crambidae* live in the stems of plants, and the distribution range is wide. So their image samples are easy to obtain (1266 images in dataset). On the contrary, *Ephydridae* have a small distribution range, and their larvae live underwater and in plant roots, so it is difficult to obtain larval images (only four images in the dataset). However, the long-tail distribution also poses challenges during model training. Categories with a large number of samples may be overfitted, while those with fewer samples might be under-represented, resulting in poor generalization. Although such a distribution mirrors real-world pest occurrence, it is preferable to minimize this imbalance for training purposes. This study attempts to reduce the impact of the long-tail effect by limiting the number of categories and narrowing the sample size gap between classes, though the effect still persists. In addition, due to the large number of differences in the classification of larvae, we chose to retain the images of larvae and do not carry out the annotation and training work for the time being. Moreover, when the damage range of too few pest species increases and image samples become easier to obtain, we will start the annotation work again to improve the dataset.

To ensure the universal applicability of the data, when collecting images, we gathered image samples from all over the world, including pest samples from China (the world’s largest rice production area), the United States (the largest production area in North America), Brazil (the largest production area in South America), Egypt (the largest production area in Africa), Thailand, South Korea, India, and other countries.

Another practical challenge is the classification of morphologically similar pests from different families. For instance, *Thripidae* and *Phlaeothripidae* exhibit similar external features, despite belonging to separate families. Under this study’s classification scheme, based on family-level taxonomy, such visually similar pests are categorized into distinct groups. While this approach aligns with taxonomic rules, the results show that both *Thripidae* and *Phlaeothripidae* achieved low classification accuracy.

The dataset is primarily composed of specimens from rice-associated pest species within the selected taxonomic families. To increase taxonomic diversity and broaden the model’s generalization ability through expanded phylogenetic coverage, some congeneric non-pest species were deliberately included. However, the taxonomic verification of these non-rice pest specimens was not the central focus of this study. As such, classification uncertainty may arise due to limited sample counts or subtle morphological differences among closely related species. Caution is advised when interpreting model outputs involving non-pest taxa in downstream applications.

No image enhancement techniques were applied to the dataset samples. In an earlier trial, we performed ten types of data augmentation, such as rotation, translation, and noise addition, on IP102, increasing the dataset to nearly 10,000 images. While the result showed an accuracy of about 96%, we had reservations about its practical significance. Although suitable augmentation can improve robustness to noise and variation, determining the most effective augmentation strategy remains a challenge that requires further careful evaluation.

As of the completion of this article (2025/3/23), the latest model in the YOLO series, YOLOv12, has only been launched for about a month. Its authors are still improving this model, so we chose the relatively stable YOLOv11 model to show the performance of our study. Once YOLOv12 is updated, we will consider using it for testing.

## 6. Conclusions

In computational pattern recognition tasks, model performance metrics (e.g., accuracy, recall, F1-score) are fundamentally governed by two interdependent factors: (1) the algorithmic efficacy of the model architecture and optimization framework, and (2) the intrinsic quality and representational adequacy of the training dataset. This dichotomy aligns with the widely accepted machine learning paradigm where model capability and data quality constitute necessary conditions for achieving optimal performance.

Our research focuses on the latter factor, the construction of a high-quality dataset for rice pest identification, addressing a critical bottleneck in agricultural computer vision applications. The quality of a dataset is operationally defined through multiple dimensions: annotation accuracy, class balance, scale diversity, ecological representativeness, and domain coverage. In specialized domains such as entomological imaging, dataset limitations frequently outweigh model architecture improvements as the primary constraint, particularly given the taxonomic complexity and morphological similarities among pest species.

In this study, we introduced a dataset focused on adult rice pests, named RP11, to address the lack of rigor and limited practical value observed in IP102. To improve classification accuracy, we classified pests in the larval stage by stage, and annotated the samples using the pest life stage most visually distinct from others, the adult stage. Additionally, we organized categories by standard pest-management taxonomic families to further improve accuracy and robustness. The resulting dataset includes 4559 adult image samples from 11 families and 2467 larval image samples from 7 families commonly associated with rice pests.

We trained the widely used deep learning model YOLOv11 on RP11. Using identical model parameters and training conditions, we conducted a comparative evaluation with IP102 over 100 iterations. The results showed that precision increased from 58.9% to 83.0%, recall from 63.1% to 79.7%, and F1-score, mAP50–95, and mAP50 improved significantly, demonstrating RP11’s clear advantage for rice pest identification tasks. This reflects the database’s higher data quality and higher use value.

## Figures and Tables

**Figure 1 life-15-00910-f001:**
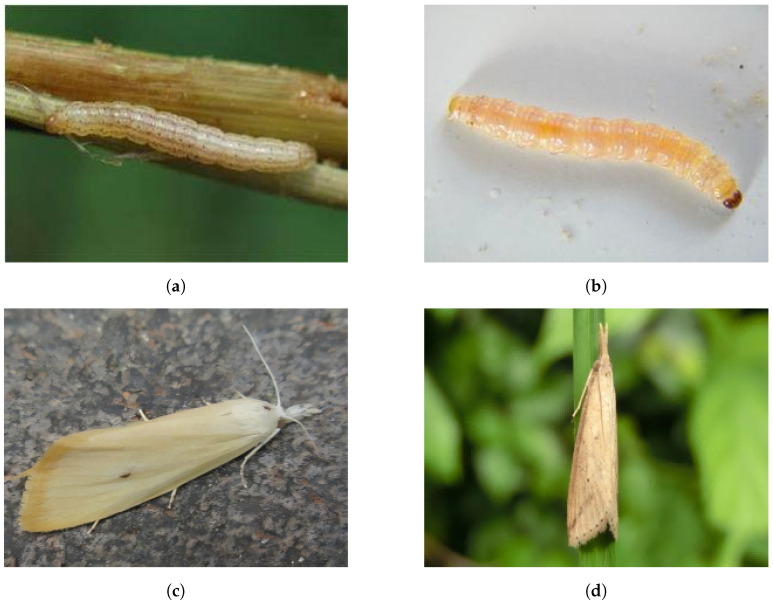
The larvae and adults of two different species: (**a**,**c**) *Chilo suppressalis* (Walker) and (**b**,**d**) *Scirpophaga incertulas* (Walker).

**Figure 2 life-15-00910-f002:**
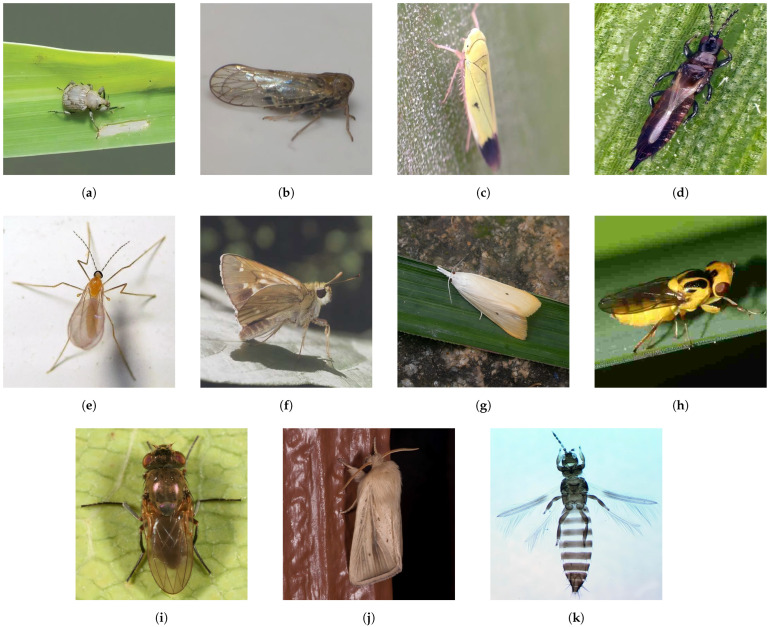
Example images from the adult categories in RP11; each image corresponds to a different family of rice pest: (**a**) *Curculionidae*; (**b**) *Delphacidae*; (**c**) *Cicadellidae*; (**d**) *Phlaeothripidae*; (**e**) *Cecidomyiidae*; (**f**) *Hesperiidae*; (**g**) *Crambidae*; (**h**) *Chloropidae*; (**i**) *Ephydridae*; (**j**) *Noctuidae*; (**k**) *Thripidae*.

**Figure 3 life-15-00910-f003:**
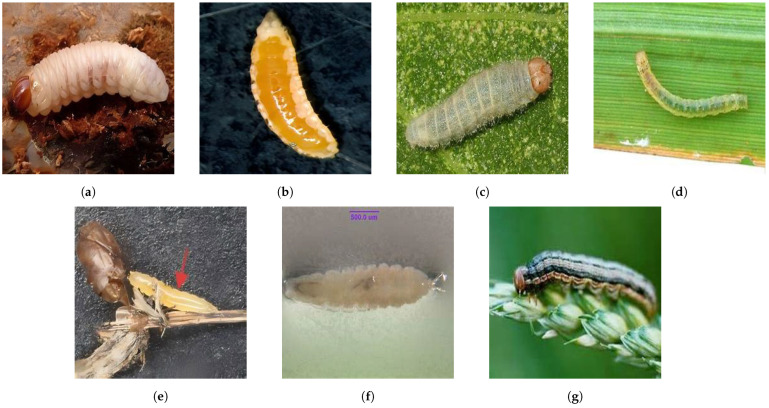
Example images from the larval categories in RP11; each image corresponds to a different family of rice pest: (**a**) *Curculionidae*; (**b**) *Cecidomyiidae*; (**c**) *Hesperiidae*; (**d**) *Crambidae*; (**e**) *Chloropidae*; (**f**) *Ephydridae*; (**g**) *Noctuidae*.

**Figure 4 life-15-00910-f004:**
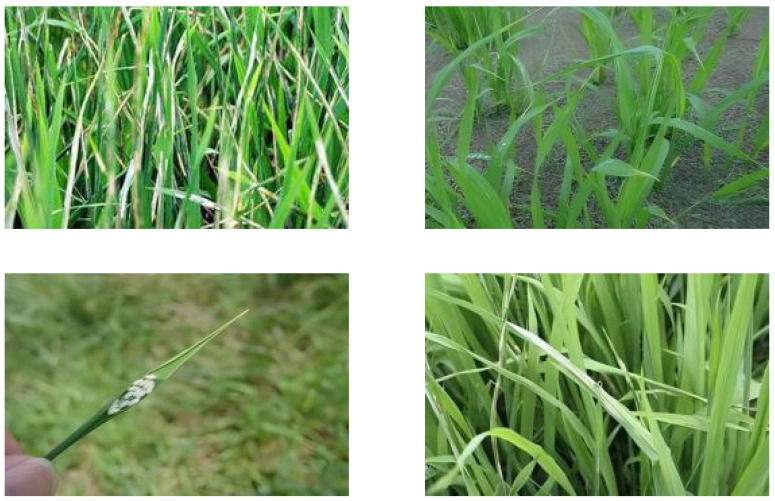
Examples of low-quality images that were filtered out.

**Figure 5 life-15-00910-f005:**
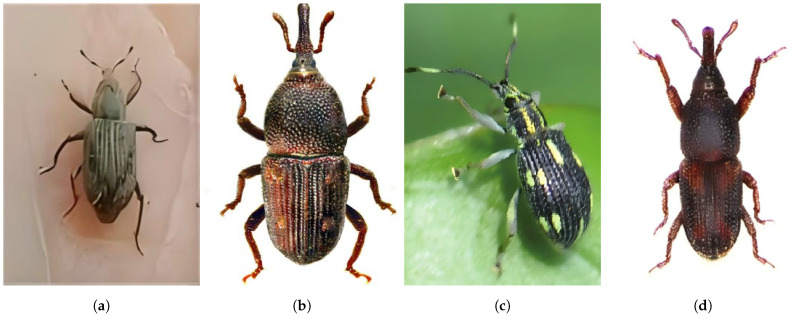
Four different species: *Curculionidae*: (**a**) *Lissorhoptrus oryzophilus*, (**b**) *Sitophilus zeamais*, (**c**) *Myllocerinus aurolineatus*, and (**d**) *Sitophilus oryzae*.

**Figure 6 life-15-00910-f006:**
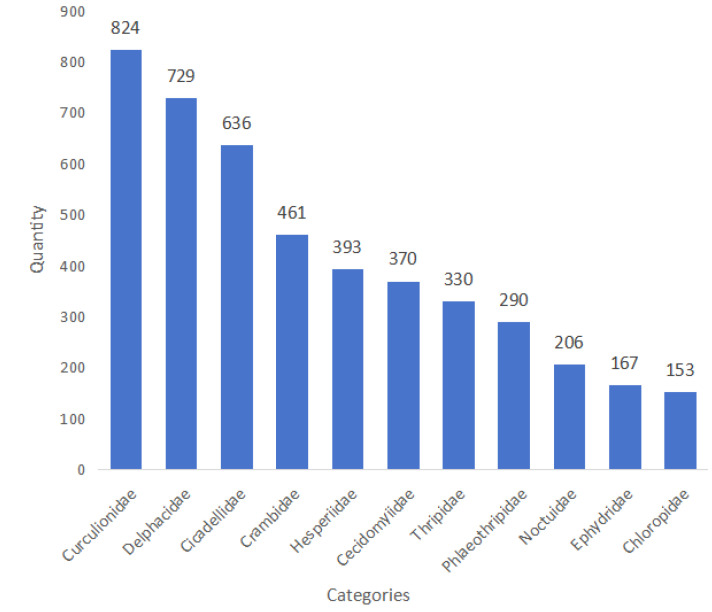
Number of samples of different adult categories in RP11.

**Figure 7 life-15-00910-f007:**
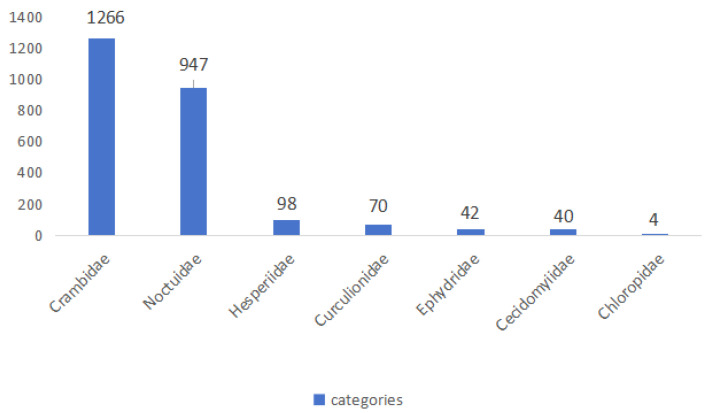
Number of samples of different larval categories in RP11.

**Figure 8 life-15-00910-f008:**
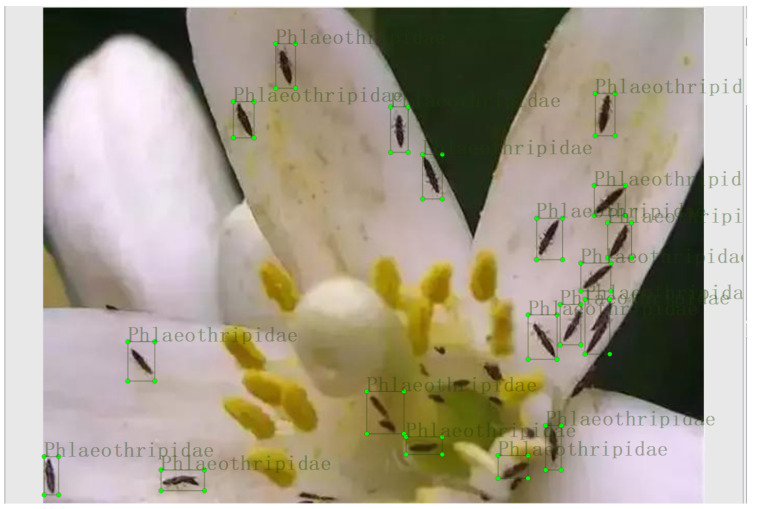
An example of an annotated result.

**Figure 9 life-15-00910-f009:**
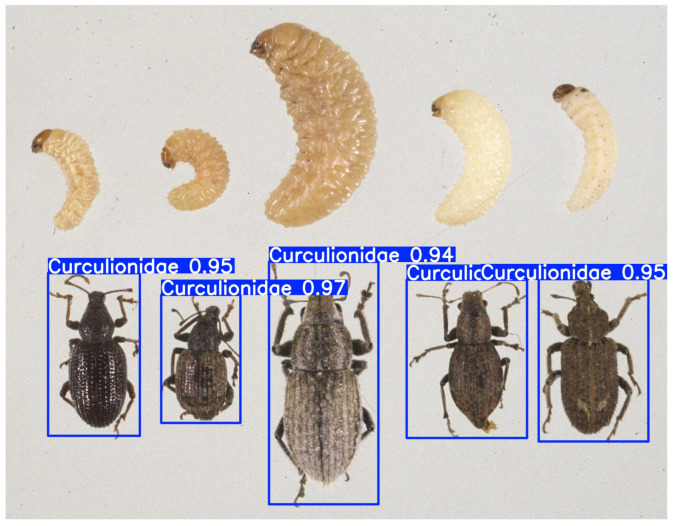
An example of a recognition result: The adult and larva of *Curculionidae*.

**Figure 10 life-15-00910-f010:**
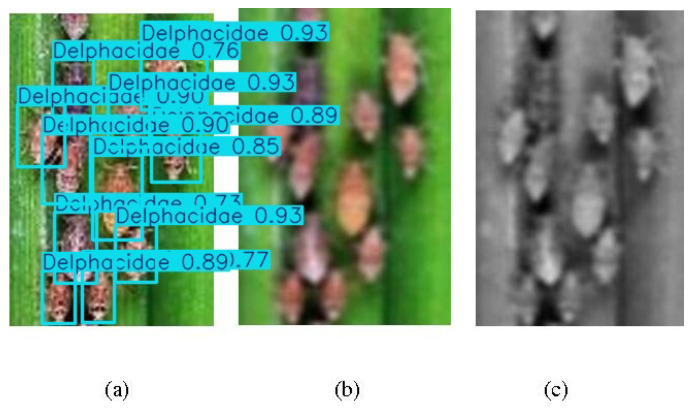
An example of a recognition result: A lot of pest instances belonging to *Delphacidae*.

**Figure 11 life-15-00910-f011:**
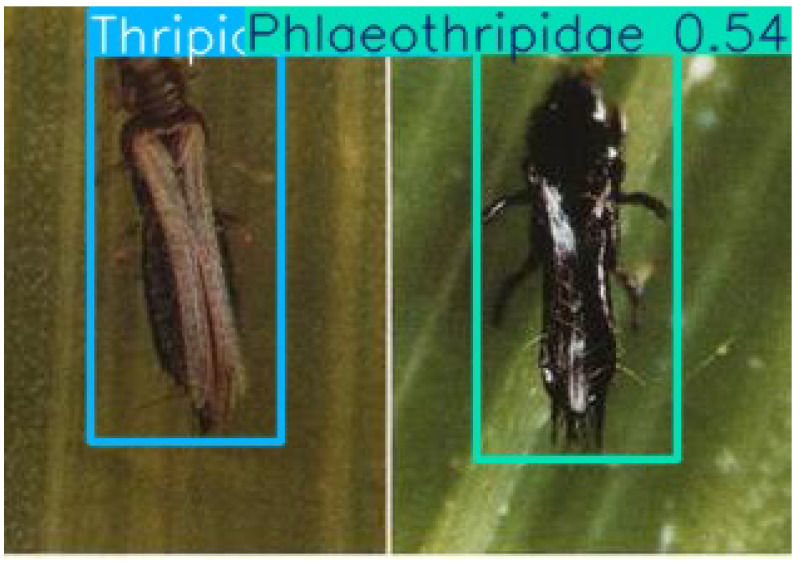
An example of a recognition result: The instance on the left is *Thripidae*, and the instance on the right is *Phlaeothripidae*.

**Figure 12 life-15-00910-f012:**
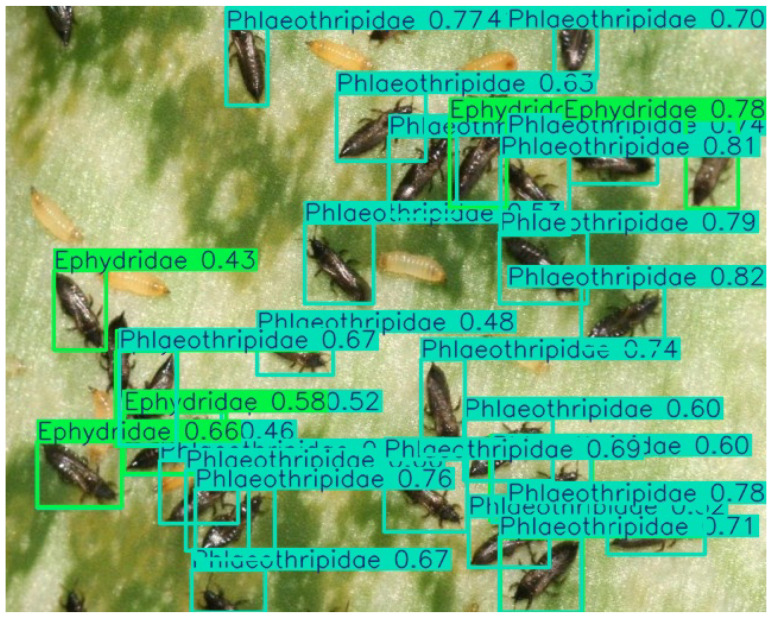
An example of a recognition result: Among a large number of samples of *Phlaeothripidae*, there are cases of incorrect detection as *Ephydridae*.

**Figure 13 life-15-00910-f013:**
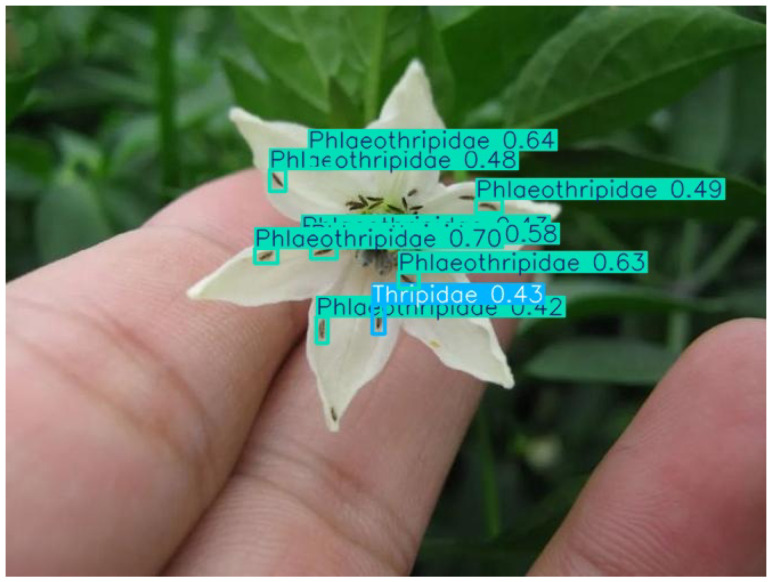
An example of a recognition result: Among a large number of samples of *Phlaeothripidae*, there are cases of missed detection or incorrect detection as *Thripidae*.

**Figure 14 life-15-00910-f014:**
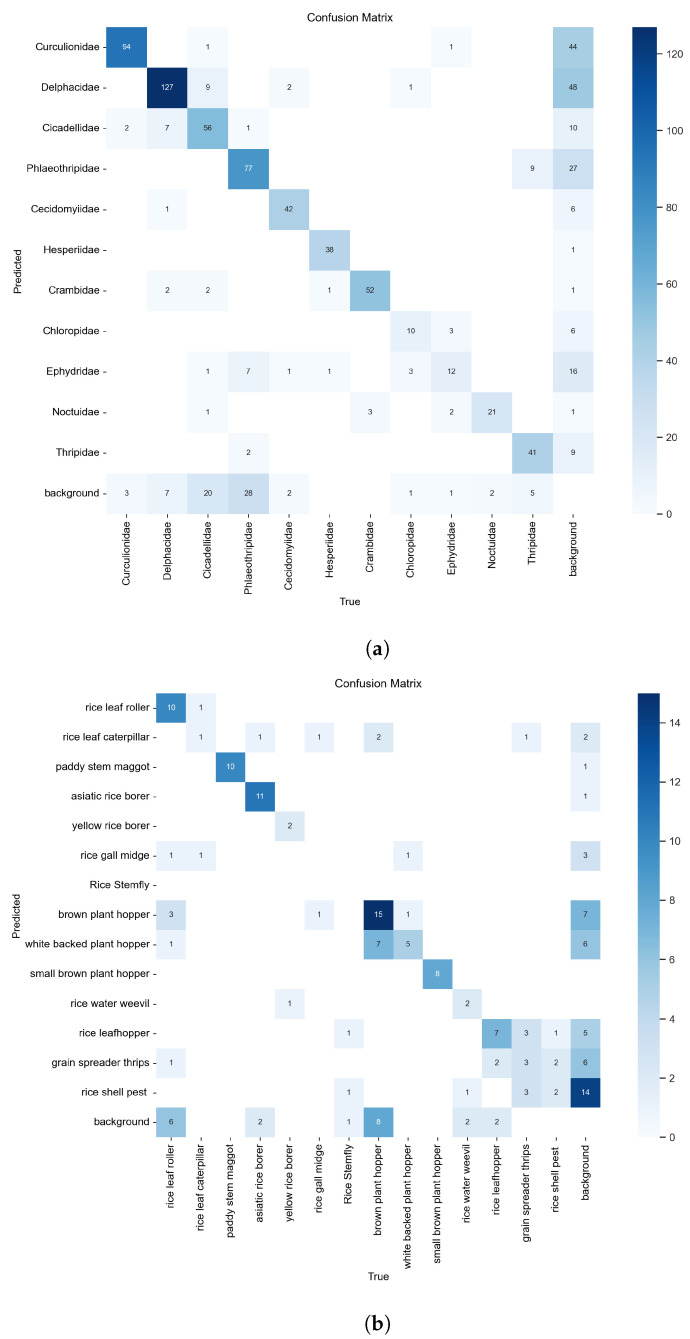
(**a**) is the confusion matrix of RP11; (**b**) is the confusion matrix of the rice pest part of IP102.

**Figure 15 life-15-00910-f015:**
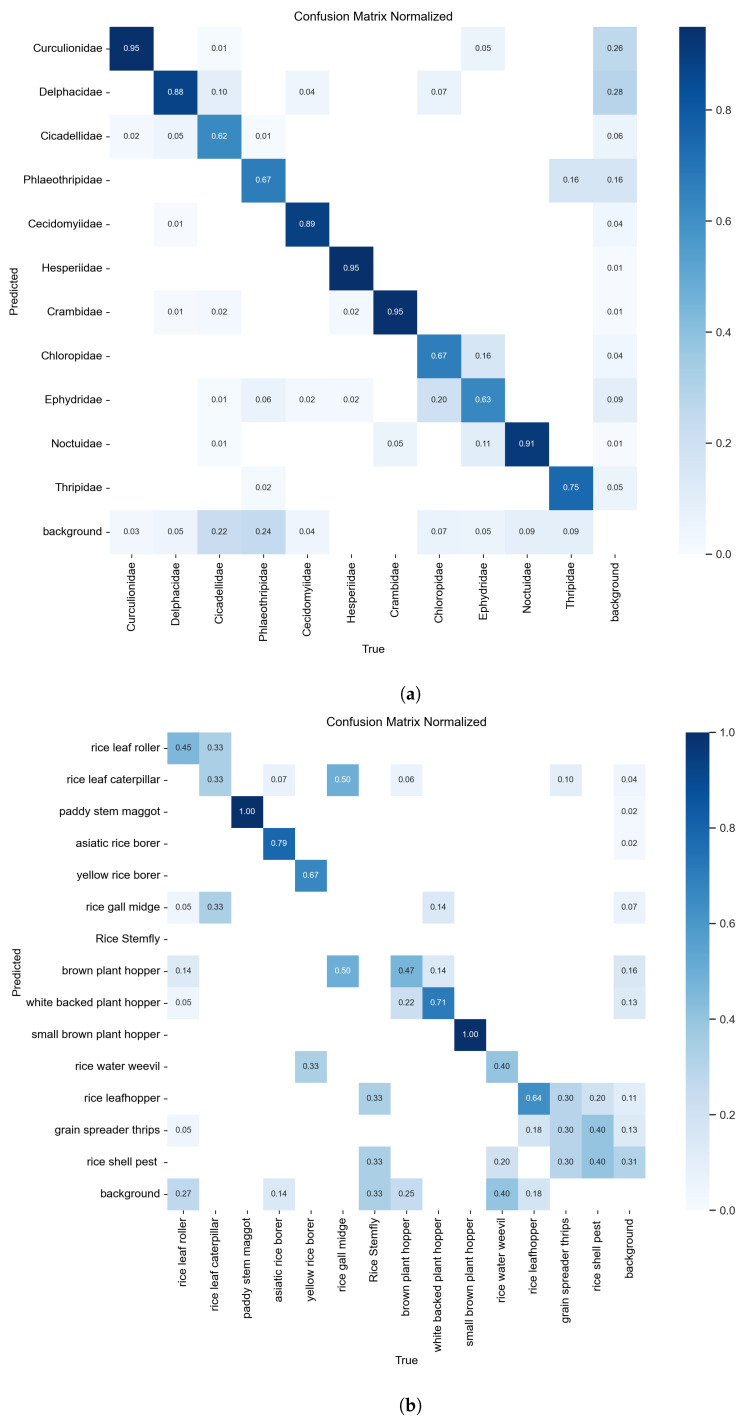
(**a**) is the normalized confusion matrix of RP11; (**b**) is the normalized confusion matrix of the rice pest part of IP102.

**Figure 16 life-15-00910-f016:**
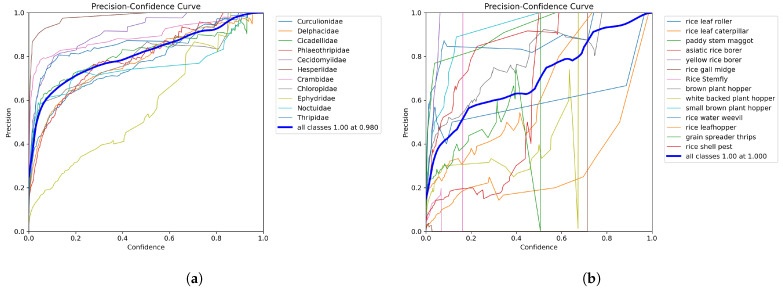
(**a**) is the precision–confidence curve of RP11; (**b**) is the precision–confidence curve of the rice pest part of IP102.

**Figure 17 life-15-00910-f017:**
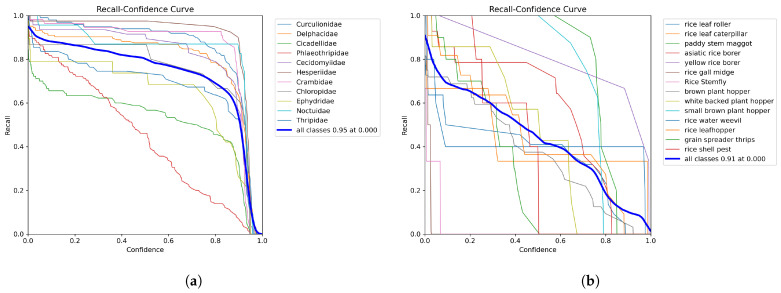
(**a**) is the recall–confidence curve of RP11, (**b**) is the recall–confidence curve of the rice pest part of IP102.

**Figure 18 life-15-00910-f018:**
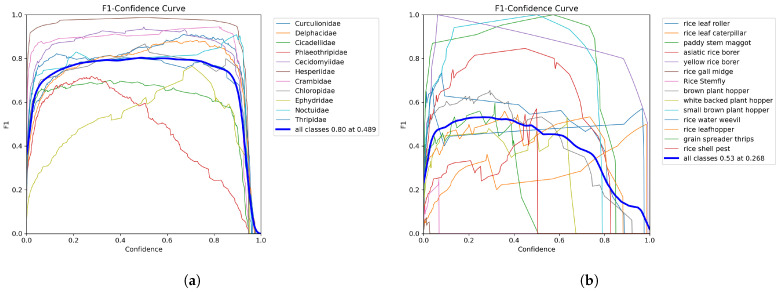
(**a**) is the F1 curve of RP11; (**b**) is the F1 curve of the rice pest part of IP102.

**Figure 19 life-15-00910-f019:**
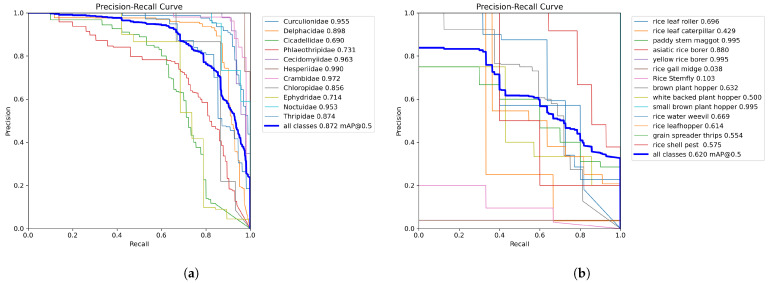
(**a**) is the precision–recall curve of RP11; (**b**) is the precision–recall curve of the rice pest part of IP102.

**Table 1 life-15-00910-t001:** The categories and their development pattern.

Developmental Pattern	Categories in RP11
**Complete metamorphosis**	*Curculionidae*
	*Cecidomyiidae*
	*Hesperiidae*
	*Crambidae*
	*Chloropidae*
	*Ephydridae*
	*Noctuidae*
**Incomplete metamorphosis**	*Delphacidae*
	*Cicadellidae*
	*Phlaeothripidae*
	*Thripidae*

**Table 2 life-15-00910-t002:** The families and species corresponding to each classification in IP102.

Family	Classification in IP102	Species
*Curculionidae*	Rice water weevil	*Lissorhoptrus oryzophilus* (Kuschel)
*Delphacidae*	Brown plant hopper	*Nilaparvata lugens* (Stål)
	Small Brown plant hopper	*Laodelphax striatellus* (Fallén)
	White backed plant hopper	*Sogatella furcifera* (Horváth)
*Cicadellidae*	Rice leaf hopper	*Nephotettix cincticeps* (Uhler)
		*Thaia rubiginosa* (Kuoh)
		*Inazuma dorsalis* (Motschulsky)
*Phlaeothripidae*	Grain spreader thrips	*Haplothrips aculeatus* (Fabricius)
*Cecidomyiidae*	Rice gall midge	*Orseoia oryzae* (Wood-Mason)
*Hesperiidae*	Rice shell pest	*Parnara guttata* (Bremer & Grey)
*Crambidae*	Yellow rice borer	*Scirpophaga incertulas* (Walker)
	Asiatic rice borer	*Chilo suppressalis* (Walker)
	Rice leaf roller	*Cnaphalocrocis medinalis* (Guenée)
	None *	*Chilo auricilius* (Dudgeno)
	None *	*Cnaphalocrocis exigua* (Butler)
*Chloropidae*	Paddy stem maggot	*Chlorops oryzae* (Matsumura)
*Ephydridae*	Rice stemfly	*Hydrellia griseola* (Fallén)
*Noctuidae*	Rice leaf caterpillar	*Naranga aenescens* (Moore)
	None *	*Sesamia inferens* (Walker)
*Thripidae*	None *	*Frankliniella intonsa* (Trybom)
	None *	*Stenchaetothrips biformis* (Bagnall)

* None means this species has images in IP102 but these images are misclassified into another category.

**Table 3 life-15-00910-t003:** The parameters of YOLOv11.

**cache**	**imgsz**	**epochs**
False	640	100
**single_cls**	**batch**	**close_mosaic**
False	8	10
**workers**	**devices**	**optimizer**
0	‘0’	‘SGD’
**amp**	**project**	**name**
True	‘run/train’	‘exp’

**Table 4 life-15-00910-t004:** Results of 9-fold cross-validation.

	Precision (%)	Recall (%)	AP50 (%)	AP50–95 (%)	F1 (%)
Exp 1	87.6	83.8	91.1	76.7	85.6
Exp 2	86.2	82.3	88.2	76.4	84.2
Exp 3	84.7	80.3	84.7	71.0	82.4
Exp 4	88.8	76.2	87.6	71.0	82.0
Exp 5	85.4	76.2	84.6	71.5	80.5
Exp 6	91.3	82.5	89.9	75.8	86.7
Exp 7	86.9	86.6	92.5	78.9	86.7
Exp 8	86.1	79.2	87.3	72.0	82.5
Exp 9	90.3	73.3	81.0	68.3	80.9

**Table 5 life-15-00910-t005:** Means, variances, and 95% confidence interval (95% CI) of k-fold cross-validation results.

	Precision (%)	Recall (%)	AP50 (%)	AP50–95 (%)	F1 (%)
Mean (%)	87.4	80.5	87.4	73.3	83.5
Variance^2^	0.0006	0.0014	0.0011	0.0011	0.0004
95% CI (%)	[85.6, 89.2]	[76.9, 84.1]	[84.7, 90.1]	[70.5, 76.1]	[81.8, 85.2]

**Table 6 life-15-00910-t006:** The number of samples allocated to each set in different categories.

Category	Training Set	Validation Set	Test Set
*Curculionidae*	659	83	83
*Delphacidae*	583	72	74
*Cicadellidae*	508	63	65
*Phlaeothripidae*	232	29	29
*Cecidomyiidae*	296	37	37
*Hesperiidae*	314	39	40
*Crambidae*	368	46	47
*Chloropidae*	122	15	16
*Ephydridae*	133	16	18
*Noctuidae*	164	20	22
*Thripidae*	264	33	33
Total	3643	452	464

**Table 7 life-15-00910-t007:** Parameters obtained by testing each classification in RP11 on the YOLOv11 model.

Class ID	Precision (%)	Recall (%)	AP50 (%)	AP50–95 (%)	F1 (%)
*Curculionidae*	78.1	93.9	95.5	82.5	85.2
*Delphacidae*	80.2	87.5	89.8	77.8	83.7
*Cicadellidae*	83.8	56.7	69.0	51.2	67.6
*Phlaeothripidae*	83.5	44.1	73.1	44.9	**57.7**
*Cecidomyiidae*	97.7	89.3	96.3	78.5	93.3
*Hesperiidae*	100.0	97.5	99.0	85.7	**98.7**
*Crambidae*	88.9	92.7	97.2	85.8	90.7
*Chloropidae*	86.3	84.4	85.6	77.7	85.3
*Ephydridae*	52.8	70.7	71.4	59.0	**60.4**
*Noctuidae*	74.9	87.0	95.3	87.9	80.5
*Thripidae*	87.1	73.5	87.4	75.8	79.7
**Total**	**Precision (%)**	**Recall (%)**	**mAP50 (%)**	**mAP50–95 (%)**	**F1-score (%)**
	83.0	79.7	87.2	73.3	81.3

**Table 8 life-15-00910-t008:** Comparison between the annotated image numbers of RP11 and IP102.

Category in RP11	Sample Number in RP11	Category in IP102	Sample Number in IP102 *
*Curculionidae*	824	Rice water weevil	155
*Delphacidae*	729	Brown plant hopper	111
		Small brown plant hopper	51
		White backed plant hopper	78
*Cicadellidae*	636	Rice leaf hopper	117
*Phlaeothripidae*	290	Grain spreader thrips **	20 **
*Cecidomyiidae*	370	Rice gall midge	93
*Hesperiidae*	393	Rice shell pest	36
*Crambidae*	461	Yellow rice borer	81
		Asiatic rice borer	166
		Rice leaf roller	174
*Chloropidae*	153	Paddy stem maggot	25
*Ephydridae*	167	Rice stemfly	28
*Noctuidae*	206	Rice leaf caterpillar	114
*Thripidae*	330	Grain spreader thrips **	20 **
**Total Quantity**	4559		1249

* Only the number of samples with annotations in IP102 is counted here. ** Category ‘Grain spreader thrips’ actually contains both categories ‘*Phlaeothripidae*’ and ‘*Thripidae*’.

**Table 9 life-15-00910-t009:** The number of samples in the three sets in RP11 and IP102.

Dataset	Training Set	Validation Set	Test Set
RP11	3643	452	464
IP102	998	125	125

**Table 10 life-15-00910-t010:** Results of precision and recall.

Dataset	Epochs	Precision (%)	Recall (%)	F1-Score (%)
RP11	100	83.0	79.7	81.3
IP102 *	100	58.9	63.1	60.9

* IP102 here refers to the rice pest part in IP102.

**Table 11 life-15-00910-t011:** Results of mAP50 and mAP50–95.

Dataset	Epochs	mAP50 (%)	mAP50–95 (%)
RP11	100	87.2	73.3
IP102 *	100	62.0	37.9

* IP102 here refers to the rice pest part in IP102.

## Data Availability

The data in this study are available on request from the corresponding author. Or download them on Kaggle: https://www.kaggle.com/datasets/dingbiao11/rp11-a-dataset-focus-on-adult-rice-pest (assessed on 25 May 2025). You can search keyword “RP11” and find this dataset with the title “RP11: A Dataset Focus on Adult Rice Pest”.

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
