# Peer review of "Improving Rice Pest Management Through RP11: A Scientifically Annotated Dataset for Adult Insect Recognition"

_life, 2025, doi:10.3390/life15060910_

Round 1
Reviewer 1 Report
Comments and Suggestions for Authors
1. It is recommended to add the open-source URL of the dataset in the abstract to increase visibility.
2. In the second paragraph of the introduction, the authors should summarize what improvements existing studies have made in identifying rice pests, rather than broadly discussing the evolution of generic object detection methods. References [12-15] are not representative enough and are not the latest research from the past two years—please introduce some more recent studies.
3. It is suggested to expand the content in the related work section and revise it in line with the first recommendation.
4. The font size in the confusion matrix is too small—please enlarge it as much as possible.
5. Regarding the experimental design, I have some suggestions. The paper improves detection model performance by augmenting the dataset. Based on the presented data, the performance gain might simply be due to the increased size of the training set. For example, if a future study further expands the 4,559 samples to 10,000, the model's accuracy will likely improve again. Thus, this process seems more like an engineering effort rather than a research contribution. The experiments do not demonstrate unique research value.
When comparing RP11 with IP102, the authors should conduct more distinctive and comprehensive validation. For instance:
(1) Use the same validation and test sets for evaluation (ensuring no data contamination).
(2) Randomly reduce the RP11 training set to match the size of IP102 to prove that RP11’s data richness is superior.
(3) Remove images in RP11 that originate from the IP102 dataset and evaluate model performance using only data collected from other sources.
These are my personal suggestions—the authors may also consider and design a more appropriate validation strategy.
6. It is recommended to include more complex object detection models for a more comprehensive comparison, such as two-stage detection methods and DETR-based detection approaches.
Author Response
请看附件(审稿人 1 的部分)。

Reviewer 2 Report
Comments and Suggestions for Authors
This manuscript presents and describes the creation and evaluation of RP11, a scientifically annotated image dataset of adult and larval rice pests. The authors propose biologically informed segmentation of pest life stages and taxonomic reorganization, aiming to improve detection accuracy using YOLOv11. RP11 is evaluated against the existing IP102 dataset to showcase classification precision and biological coherence improvements. While the idea is relevant and potentially valuable for agricultural entomology and deep learning, the execution of the work suffers from substantial structural, scientific, and methodological shortcomings that limit its impact and reliability.
- Lines 1–15: The abstract overclaims performance improvement. Add actual numeric metrics. In addition, vague statements like "outperforms IP102" should be replaced with statistical significance values.
- Lines 16–71: The introduction is too long and contains redundant descriptions of the YOLO series. In this regard, cut or move YOLOv3-v12 historical details (Lines 39–51) to the M&M section.
- Lines 72–104: The section "related work" should be part of the introduction. However, it is repetitive and lacks critical assessment. Add a paragraph contrasting the novelty of RP11 vs IPR16/IP_RicePest/IP102.
- M&M section: Add a full table of class distribution. In addition, clarify if the YOLO anchor box adaptation was performed. Add cross-validation protocols if applicable.
- Taxonomic classification is presented at the family level, ignoring large morphological and ecological variance within families. For example, labeling Curculionidae or Crambidae generically introduces confusion for downstream users. Identification at the species or genus level is more meaningful for field application. The manuscript must mention, explain, and discuss these plausible limitations.
- Lines 193–205: Inclusion of non-pest species from the same family under the same class label (e.g., Sitophilus oryzae grouped with Lissorhoptrus oryzophilus) introduces semantic noise and misrepresents target pest control scenarios. Explanations and justifications must be provided.
- Lines 246–251: The annotation method is described as YOLO-compliant, yet there is no inter-annotator validation, error-checking process, or quality assurance procedure described. No mention of bounding box overlap rate or false-positive handling, which should be mentioned and explained in the manuscript.
- Lines 167–205: Web-scraped image sources are described vaguely. No licensing repository details, scraping protocol, or dataset cleaning strategy are appropriately cited. In addition, no visual inspection statistics or example rejected samples are provided. These considerations must be included in the manuscript.
- Lines 280–285, Table 4: YOLOv11 is compared against IP102 using non-identical class distributions and vastly different sample sizes, invalidating fair performance comparison.
- Lines 312–316: The training/test split of 8:1:1 is acceptable but not stratified by class, exacerbating long-tail imbalance and potential overfitting in common classes.
- Lines 296–306: The model's inability to detect larvae is acknowledged but dismissed as a strength. This is contradictory and reduces the model's utility in real-world applications.
- Figures 8–10: Only anecdotal recognition outputs are shown. No failure cases are presented.
- Statistics, data quality, and interpretation: To enhance the scientific rigor and transparency of the dataset, the authors should provide complete metadata tables, detailed licensing information, and a clearly defined image cleaning and curation protocol. The manuscript must include appropriate statistical analyses to support performance comparisons, specifically, significance testing (e.g., p-values or confidence intervals) to validate claims of superiority over existing datasets. Furthermore, the current model evaluation lacks robustness due to evident dataset imbalance; this should be addressed using class rebalancing strategies such as data augmentation, stratified sampling, or weighting. The annotation pipeline also requires improvement through a formal quality assurance process, including inter-annotator agreement metrics or error rate reporting. Finally, model evaluation should be expanded beyond anecdotal examples to include comprehensive error analysis, highlighting both false positives and false negatives, to offer a more realistic assessment of model reliability.
- Lines 497–499: The Kaggle link is mentioned casually. No DOI, metadata standards, or FAIR (Findability, Accessibility, Interoperability, Reusability) principles are applied.
- Results section: Table 4 and Figures 13–16 should be supplemented with standard deviation, confidence intervals, or p-values for comparisons. The long-tail distribution is admitted but not mitigated in any meaningful way.
- Lines 407–456: The discussion is mostly descriptive. There is no critical interpretation of why certain classes perform poorly. In addition, no ablation study or experiment justifies taxonomic restructuring.
- In conclusion, while RP11 as a concept has merit, this manuscript overstates its contribution, lacks biological rigor, and does not meet the reproducibility and transparency standards of a scientific dataset paper.
The manuscript is mostly grammatically correct but overly verbose and passive. Additionally, technical terms are inconsistently used. For example, "Recall" is used in multiple senses, and should be clarified. Sentences like "This can be attributed to..." (Line 293) are speculative without evidence. Detailed scrutiny throughout the manuscript is recommended to revise these and additional grammar and stylistic issues.
Author Response
Please see the attachment(the part of reviewer 2).

Reviewer 3 Report
Comments and Suggestions for Authors
The paper “Improving Rice Pest Management Through RP11: A Scientifically Annotated Dataset for Adult Insect Recognition” represents a meaningful step toward enhancing pest detection in rice agriculture through deep learning, specifically by refining the broader IP102 dataset into a more targeted RP11 dataset. While this effort is commendable, several important issues need to be addressed before the manuscript can be considered for acceptance.
Comments:
- The inclusion of web-crawled images expands the dataset, but may also introduce mislabeled or low-quality entries. It’s essential for the authors to clarify whether these images were validated by qualified entomologists.
- Were trained taxonomists or domain experts involved in the image reannotation process and in assigning accurate scientific names?
- The methodology for distinguishing larvae from adult insects is unclear. Given that some larval forms can closely resemble other developmental stages or species, this aspect warrants further explanation.
- Lines 124–130 should be rewritten for better clarity.
- Even with a larger overall dataset, larval images (2,467) remain significantly underrepresented compared to adult images (4,559). This imbalance could affect model performance.
- The authors should discuss whether class imbalance poses a challenge in the RP11 dataset, and specify the resolution and quality range of the images sourced via web crawling.
- Since the dataset appears optimized for YOLO, this might limit its adaptability to other models like DETR or EfficientDet. Broader architectural compatibility should be considered.
- The evaluation methodology does not mention k-fold cross-validation, a common technique to ensure model robustness and generalizability.
- What was the rationale for selecting YOLOv11 over newer versions like YOLOv8 or other popular architectures such as Faster R-CNN and EfficientDet? A comparative discussion would help clarify this choice.
- It is unclear whether the dataset encompasses diverse ecological zones. A geographically narrow dataset may hinder the model’s effectiveness in varied real-world environments.
- IP102 is the only baseline used. Including comparisons with other rice pest datasets, such as Paddy Doctor or RPD2021, would strengthen the study.
- The paper does not incorporate explainable AI tools like Grad-CAM or SHAP, which can provide insights into how the model is making decisions. Their use would enhance transparency.
- The manuscript lacks details about inter-annotator agreement or quality control protocols, especially important for the more ambiguous larval annotations.
- Were statistical significance tests conducted to confirm the performance differences observed in the study? This information is crucial for validating results.
- Since visual differentiation between larval and adult stages can be biologically ambiguous, the authors should elaborate on any biological verification methods used to support classification accuracy.
- Lastly, did the authors apply any explainability methods (e.g., Grad-CAM) to evaluate model decisions? And what is the comparative computational cost of training YOLOv11 on RP11 versus IP102?
Author Response
Please see the attachment(the part of reviewer 3).

Reviewer 4 Report
Comments and Suggestions for Authors
For the manuscript by Ding et al "Improving Rice Pest Management through RP11":
Having worked on pest identification through pictures of stages, I appreciate very much the improvements in identification of rice pests made by this research. Consequently, I have only a few minor suggestions for improvement.
- Figure 8 needs to be made larger. Alternatively, can there be an ability to greatly expand the picture by clicking on a corner so we can actually see the data more clearly?
- Line 416: “To take another example, the …”
Recommendation: Accept (more or less) as is.
Author Response
Please see the attachment(the part of reviewer 4).

Round 2
Reviewer 1 Report
Comments and Suggestions for Authors
The authors have revised the introduction, related work, and methodology sections of the manuscript, resulting in improved quality and rigor in these parts. Overall, the revisions are well done. However, I noticed that some comments in the authors' response letter were not adequately addressed, which might cause confusion for readers. Please carefully review the responses again, provide thorough explanations, or include supporting data where needed. Once these improvements are completed, I believe the manuscript will meet the publication standards.
Author Response
Thank you for pointing out our problem. For specific explanations, please refer to the attachment.

Reviewer 2 Report
Comments and Suggestions for Authors
The authors have adequately addressed my comments, and I am satisfied with their responses; therefore, the manuscript can now be considered for publication.
Author Response
Thank you for your recognition of our work.
Reviewer 3 Report
Comments and Suggestions for Authors
The authors made substantial changes in the manuscript.
Author Response

(The authors gave the same response as above.)
